

# Experimental removal of introduced slider turtles offers new insight into competition with a native, threatened turtle

Max R. Lambert[1], Jennifer M. McKenzie[2], Robyn M. Screen[3], Adam G. Clause[4], Benjamin B. Johnson[5], Genevieve G. Mount[6], H Bradley Shaffer[7] and Gregory B. Pauly[4]

[1] Department of Environmental Science, Policy, and Management, UC Berkeley, Berkeley, CA, United States of America
[2] Department of Forestry and Natural Resources, University of Kentucky, Lexington, KY, United States of America
[3] Department of Biology, University of Hawaii at Manoa, Honolulu, HI, United States of America
[4] Urban Nature Research Center & Section of Herpetology, Natural History Musem of Los Angeles County, Los Angeles, CA, United States of America
[5] Department of Ecology and Evolutionary Biology, Cornell University, Ithaca, NY, United States of America
[6] Biology Department, Louisiana State University, Baton Rouge, LA, United State of America
[7] Department of Ecology and Evolutionary Biology & La Kretz Center for California Conservation Science, University of California, Los Angeles, Los Angeles, CA, United States of America

Corresponding author
Max R. Lambert, mrl24@berkeley.edu

## ABSTRACT

The red-eared slider turtle (*Trachemys scripta elegans*; RES) is often considered one of the world's most invasive species. Results from laboratory and mesocosm experiments suggest that introduced RES outcompete native turtles for key ecological resources, but such experiments can overestimate the strength of competition. We report on the first field experiment with a wild turtle community, involving introduced RES and a declining native species of conservation concern, the western pond turtle (*Emys marmorata*; WPT). Using a before/after experimental design, we show that after removing most of an introduced RES population, the remaining RES dramatically shifted their spatial basking distribution in a manner consistent with strong intraspecific competition. WPT also altered their spatial basking distribution after the RES removal, but in ways inconsistent with strong interspecific competition. However, we documented reduced levels of WPT basking post-removal, which may reflect a behavioral shift attributable to the lower density of the turtle community. WPT body condition also increased after we removed RES, consistent with either indirect or direct competition between WPT and RES and providing the first evidence that RES can compete with a native turtle in the wild. We conclude that the negative impacts on WPT basking by RES in natural contexts are more limited than suggested by experiments with captive turtles, although wild WPT do appear to compete for food with introduced RES. Our results highlight the importance of manipulative field experiments when studying biological invasions, and the potential value of RES removal as a management strategy for WPT.

## INTRODUCTION

The International Union for Conservation of Nature (IUCN) has labeled the red-eared slider turtle (*Trachemys scripta elegans*; RES) one of the "world's worst invasive species" (*Lowe et al., 2000*). RES are native to the central United States—from West Virginia through Texas to eastern New Mexico and north into Illinois—but are now established throughout much of the country. RES are also globally distributed, having been released on every continent except Antarctica with robust populations in Europe and Asia. These introductions predominantly result from the release of unwanted pet turtles (*Kraus, 2009*; *Rhodin et al., 2017*). Results from laboratory and mesocosm experiments suggest that RES can outcompete native European and eastern North American freshwater turtles for food and basking sites (*Cadi & Joly, 2003*; *Cadi & Joly, 2004*; *Polo-Cavia, Lopez & Martin, 2010*; *Polo-Cavia, Lopez & Martin, 2011*; *Pearson, Avery & Spotila, 2015*). While such controlled experiments are informative, they can also inflate the effects of competition compared to *in situ* field manipulations (*Skelly, 2002*; *Winkler & Van Buskirk, 2012*). Comparing laboratory and mesocosm experiments with field manipulations is a critical step to a more complete understanding of the strength and mechanisms underlying species interactions in nature. However, to our knowledge, no study has experimentally tested for competition between non-native RES and any native turtle species in the wild.

Basking sites are a key resource for thermoregulation, disease control, and reproduction in freshwater turtles (*Ernst & Lovich, 2009*), and previous *ex situ* experiments suggest that basking sites are an important axis of competition between native turtles and introduced RES (*Cadi & Joly, 2003*; *Polo-Cavia, Lopez & Martin, 2010*). Prior work in the University of California, Davis Arboretum waterway (hereafter, UCD Arboretum) found that introduced RES and native western pond turtles (*Emys marmorata*; WPT) sometimes bask at the same sites (Fig. 1), although they tend to use basking sites that differ physically and spatially (*Lambert et al., 2013*). In particular, WPT and RES predominantly bask in the western and eastern ends of the UCD Arboretum, respectively. While WPT basking is not related to particular habitat characteristics, RES basking is related to sites with more human activity, steel mesh basking substrates, deeper water, and shallower slopes. Whether these basking site differences are the result of species-specific habitat choices or competition has never been resolved and requires an experimental approach.

Many freshwater turtles, including both WPT and RES, are dietary generalists as adults and consume a broad array of food items, though they tend to shift from higher rates of carnivory when young to higher rates of herbivory as adults (*Ernst & Lovich, 2009*). Even so, laboratory and mesocosm experiments suggest RES might directly interfere with native turtle food consumption through aggressive behaviors or higher food consumption rates, which can limit food availability and growth rates of less competitive turtles (*Cadi & Joly, 2004*; *Polo-Cavia, Lopez & Martin, 2011*; *Pearson, Avery & Spotila, 2015*). Additionally, if turtle densities are high for a given habitat, exploitative competition could limit food availability, both intra- and interspecifically, and therefore decrease growth rates and / or body condition of native species.

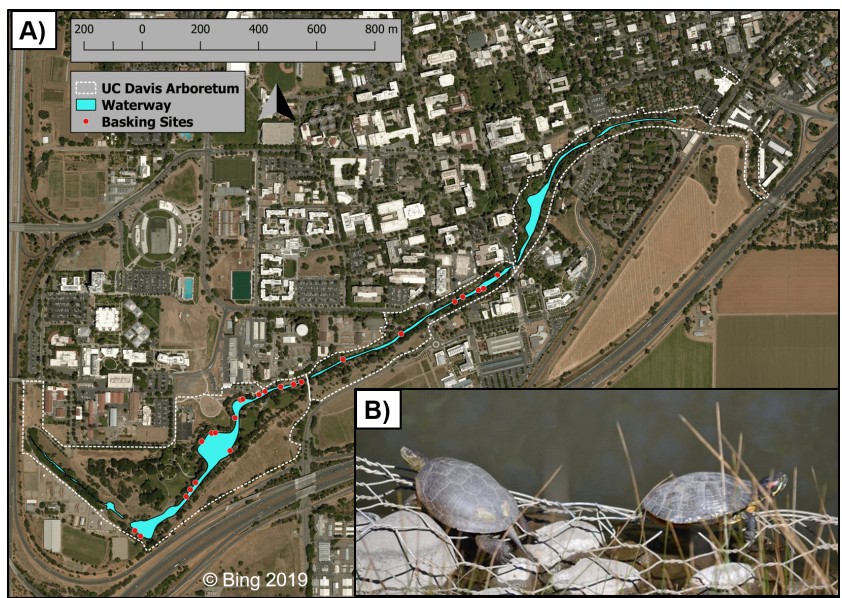

**Figure 1** **The UC Davis Arboretum waterway, turtle basking sites, and basking turtles.** Map (A) of the UC Davis Arboretum outlined with a dashed white line with the waterway in blue and turtle basking sites displayed as white-rimmed red circles. Seen basking (B) are a native western pond turtle and an introduced red-eared slider side-by-side in the Arboretum. Map data © Bing 2019. Photo credit Max Lambert.

Here, we present the results of an *in situ* field experiment where we substantially reduced the introduced RES population at the UCD Arboretum to test for competition with WPT. Because the waterway is disjunct from other turtle populations, the UCD Arboretum represents a closed system for WPT that is well suited for experimental manipulations; natural immigration/emigration is not possible for freshwater turtles in this system, although occasional human-assisted transport does occur, particularly with RES released into the waterway. Our experiment is the first to explicitly test whether invasive species removal, a commonly-advocated management practice for invasive species including RES (*Gaeta et al., 2015*; *García-Díaz et al., 2017*), influences the basking behavior and body condition of a native turtle in the wild. If the distribution of WPT basking is a result of direct, competitive exclusion by RES from optimal basking sites, then RES removal should result in an increase of post-removal WPT basking at sites previously dominated by RES. Alternatively, if WPT basking activity does not significantly change in this manner after RES removal, then existing behavioral basking differences between the two species likely reflect species-specific habitat preferences, competitive superiority of WPT, or both. We also assessed WPT body condition pre- and post-removal as a proxy for whether removing RES improves WPT access to food resources. If introduced RES compete with WPT for food, then removing RES should result in an increase in WPT body condition. Given the broad overlap of these two species across California (*Thomson, Spinks & Shaffer, 2010*; *Thomson, Wright & Shaffer, 2016*), the range-wide imperilment of WPT (*Spinks et al., 2003*; *Thomson, Wright & Shaffer, 2016*), and the current Status Review for possible WPT listing

under the U.S. Endangered Species Act (*USFWS, 2015*), this experiment is directly relevant to ongoing WPT management actions.

## METHODS

UC Davis IACUC Protocols #15263 and #16227 and California Department of Fish and Wildlife Scientific Collecting Permits #2480, #4307, and #11663 approved this work. We conducted all analyses in the R statistical language (version 3.5.2).

### Study site

Our study took place at the UCD Arboretum (38.53, −121.76), a permanent waterway extending along the southern border of the UC Davis campus, Yolo County, California, USA (Fig. 1). The UCD Arboretum was formed in the 1870s when the historical north fork channel of Putah Creek was diverted into the south fork (*Larkey, 1980*). This waterway is 2.4 km long, ca. 4 ha in surface area, and averages 15 m wide and 1 m deep (*Spinks et al., 2003*). Terrestrial habitat surrounding the waterway is irrigated and landscaped with predominantly non-native vegetation (*Spinks et al., 2003*). A 1.5–2.5 m wide paved path encircles the entire waterway within 5–10 m of the water's edge. This path is regularly used by pedestrians, bicyclists, and maintenance vehicles which influence turtle basking (*Lambert et al., 2013*; *Costa, 2014*). The waterway's shoreline—including basking sites—is a combination of concrete, exposed dirt, and landscaping steel mesh which has been exposed by erosion (*Lambert et al., 2013*).

### Turtle trapping and RES removal

Across the UCD Arboretum, we deployed baited submersible traps in optimal habitat for both RES and WPT over approximately 900 trap-days from 10 July–1 August, 2011 and again from 13–29 September, 2011. We supplemented this trapping with dip netting and opportunistic hand captures during both periods, and with a fyke net and a basking trap during the latter period. Dip netting and hand captures were targeted at RES but other trapping was not. We removed and euthanized all RES, depositing most specimens at the UC Davis School of Veterinary Medicine, the Natural History Museum of Los Angeles County, or the UC Davis Museum of Wildlife and Fish Biology. Beginning in the 1996, our group uniquely marked each captured WPT with scute notches using a handheld file or (juveniles only) nail clippers (*Spinks et al., 2003*). We similarly marked any new WPT during this trapping effort. We used linear regression to test whether our trapping depleted the RES population over time by regressing cumulative RES captures against trapping day for adult RES (*Krebs, 1989*). Using likelihood ratio tests (*Crawley, 2013*) to assess model fits, we compared a quadratic model, which would indicate population depletion, to a linear model, which would indicate that the RES population was not leveling off with our removal effort.

### WPT body condition

To estimate changes in WPT body condition, we trapped for one week the year following RES removal, from 27 May–2 June, 2012. Due to logistical constraints we were unable to

trap later in the summer at a similar time as in 2011. Differences in trapping dates may influence body condition analyses because females may be gravid and therefore heavier in the earlier 2012 sampling or because all turtles may have had more time to put on mass during the later 2011 sampling. However, these effects are likely limited. In both 2011 and 2012 we measured WPT plastron length (notch-to-notch; mm) with dial calipers and body mass (g) with Ohaus CS2000 digital pan scales (*Iverson & Lewis, 2018*). We used a linear mixed-effects model (function 'lmer', R package "lme4"; *Bates et al., 2015*) to test whether WPT body condition (i.e., differences in mass controlling for body length) changed after the RES removal (*Cadi & Joly, 2003*; *Schulte-Hostedde et al., 2005*; *Litzgus, Bolton & Schulte-Hostedde, 2008*). Our model of WPT mass controlled for plastron length and included treatment (pre- or post-removal) and sex as fixed effects and individual WPT as a random effect to control for repeated measures. This model simultaneously regresses body mass against plastron length and tests for differences in the residuals of this model between study year and sex. We used likelihood ratio tests to assess the significance ($\alpha < 0.05$) of fixed effects and removed non-significant variables from our model. We obtained full model conditional $R^2$ ($cR^2$) for fixed and random effects combined and a marginal $R^2$ ($mR^2$) for the model's fixed effects alone (function 'r.squaredGLMM', package "MuMIn", *Barton, 2018*).

## Basking site monitoring

We conducted binocular surveys of 24 pre-selected basking sites (Fig. 1) for 34 total days—16 days pre-removal between 18 March and 22 April 2010 (*Lambert et al., 2013*) and 18 days post-removal between 18 March and 22 April 2012. Following *Lambert et al. (2013)*, we performed all surveys between 1,000 and 1,500 hr to coincide with the expected maximum turtle basking activity during this time of year. We surveyed all sites once daily in rapid succession to avoid counting the same turtle at multiple sites. Each survey was performed in under one hour, at a distance of ca. 10–100 m from turtles, and did not noticeably disturb basking turtles. MRL and S. Nielsen conducted basking surveys in 2011 and JMM and RMS conducted 2012 surveys; all surveyors were trained by GBP and HBS. During each survey we recorded the number of individuals of each species basking at each basking site as well as water temperature because we previously found that basking activity of both species increases more with warmer water temperatures than with air temperatures (*Lambert et al., 2013*). We also obtained air temperature data from the UC Davis Russell Ranch Weather Station which is located ca. 4 km NW of the UCD Arboretum.

## Modeling the effects of RES removal on turtle basking

We tested for changes in the relative basking distribution of WPT and RES (i.e., the proportion of basking turtles that were Emys − Emys / (Emys + Trachemys) across the UCD Arboretum pre- and post-RES removal using a generalized linear mixed effects model (GLMM) with a binomial family for proportion data (function 'glmer', R package "lme4"). A binomial GLMM accounts for binary data (two species here) and variation in sample sizes across basking sites and survey days. We modeled WPT:RES basking as a function of treatment (pre- or post-removal) and the distance of each basking site from the west end

of the UCD Arboretum because turtle basking distributions were previously shown to vary west-east (*Lambert et al., 2013*). We accounted for repeated measures by treating survey date as a random effect (*Lambert et al., 2013*). To explore site-specific changes in the ratio of the two species, we also used individual binomial GLMMs for each basking site.

In addition, we modeled the absolute basking abundance of both species pre- and post-removal using Poisson GLMMs (function 'glmer', R package "lme4") for count data. Our approach here was the same as with the binomial GLMM and, if an interaction between treatment and distance from the west end was significant, we used individual GLMMs for each year to test the pattern and strength of turtle basking distributions across the UCD Arboretum in each year. To test whether certain basking sites made up larger or smaller proportions of total WPT basking observations pre- or post-removal, we used contingency tables, focusing on the five most heavily-used turtle basking sites (combined for both species) pre-removal (sites P, O, E, Q, and R) and site X, the most heavily-used turtle basking site post-removal.

## RESULTS

### Trapping and RES removal

We removed and euthanized 177 RES (100.6 kg total biomass), including 28 adult males (16.3 kg), 72 adult females (79.4 kg), and 77 juveniles (4.9 kg, defined as $\leq 100$ mm carapace length; *Ernst & Lovich, 2009*). A quadratic (rather than linear) model fit our data best (likelihood ratio test $p < 0.0001$, full model $R^2 = 0.95$) and showed RES captures leveling off, signifying we had removed a substantial fraction of the RES population.

We also captured, marked, and released 115 unique WPT (62.7 kg total biomass) comprising 51 males (36.1 kg), 36 females (24.1 kg), and 28 juveniles (2.5 kg, defined as $\leq 110$ mm plastron length; *Holland, 1991*).

### WPT body condition

While we trapped a larger number of WPT in each year, we trapped 25 unique adult WPT in both 2011 and 2012; we used these 25 WPT for the body condition analysis. The body condition linear model showed no interaction between treatment and sex ($p = 0.92$) and so we removed this interaction from the model. Sex ($p = 0.009$), treatment ($p < 0.001$), and plastron length ($p < 0.0001$) were significant (full model $cR^2 = 0.95$, $mR^2 = 0.86$). For a given plastron length, males were on average 61.54 g ($\pm23.55$ SD g) heavier than females. Although the 25 WPT measured before and after RES removal showed individual variation in their degree of body condition change post-removal (Fig. 2A), on average they were 39.80 g ($\pm9.92$ SD g) heavier for a given plastron length post-removal (Fig. 2B).

### Basking site monitoring

We recorded 283 WPT and 645 RES observations in 2010 but only 43 WPT and 61 RES observations in 2012. Although the reduction in numbers of observed WPT was unexpected, we do not believe this reflects a decline in the WPT population. From 27 May–2 June 2012, we trapped 54 unique WPT over seven days and a Schnabel multiple capture-mark-recapture population estimate (*Krebs, 1989*) derived from trapping data

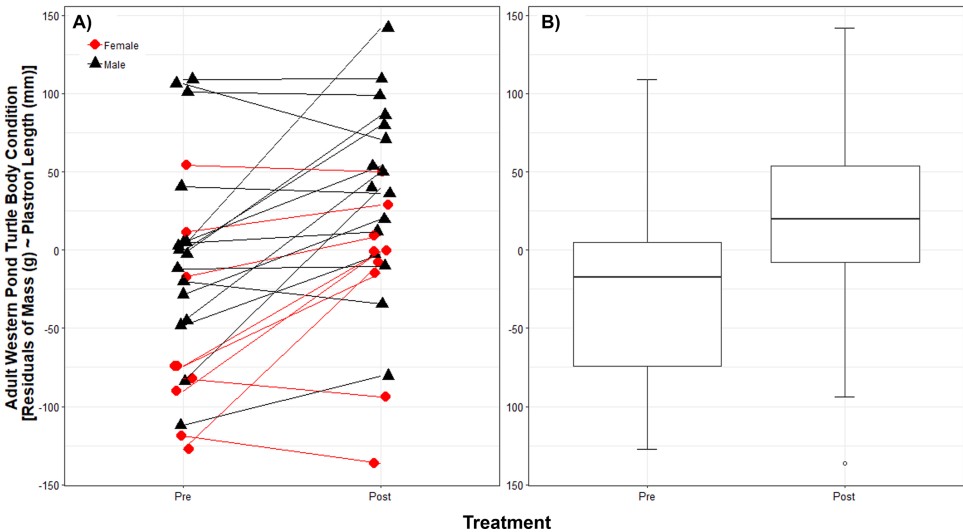

**Figure 2** **Native western pond turtle (WPT) body condition before and after introduced red-eared slider (RES) removal.** Body condition is shown as the residuals of body mass regressed against plastron length. Individual WPT varied in their body condition response to introduced RES removal (A) but body condition generally improved. On average (B) WPT are 39.80 g heavier after RES removal. Boxplot hinges show the 25th and 75th body condition percentiles, whiskers show the extent of data within 1.5 times the interquartile range, and the center line is the median for each treatment year pre- and post-removal.

(beginning in the mid 2000s) suggest that ca. 162 WPT (including 10 newly-marked juveniles) were present in the UCD Arboretum immediately after our post-removal surveys (J McKenzie, R Screen, and G Pauly, pers. comm., 2014); this estimate is similar to pre-removal estimates of WPT population size (ca.146 WPT, including 18 newly-marked juveniles). Given these estimates, we are confident that the WPT population was essentially unchanged during our experiment, and thus our focus on the relative basking distributions of turtles at monitored basking sites meaningfully reflects the impact of our removal experiment and not a catastrophic decline in WPT.

The basking sites most commonly used by WPT pre-removal were generally the same sites used post-removal (Figs. 3A, 3B). We recorded WPT basking at 15 of 24 basking sites pre-removal, but at only eight of 24 sites post-removal. WPT were absent from 8 sites they used pre-removal (although of these, only two were frequently used pre-removal: sites A and N) and were present at one additional site where they were not recorded pre-removal (site B). We recorded RES basking at 17 of 24 basking sites pre-removal, and only eight of 24 sites post-removal (Figs. 3A, 3B). RES were absent from nine sites they used pre-removal and were not recorded using new sites post-removal.

Water temperatures were warmer in 2010 (17.0 C $\pm$ 1.71 SD) than in 2012 (15.4 C $\pm$ 2.12 SD; two-tailed $t$-test, $p < 0.0001$). However, maximum daily air temperatures (averaged across all days of each survey period) were not different between years (2010, 19.2 C $\pm$ 4.13 SD; 2012, 18.8 C $\pm$ 5.25 SD; two-tailed $t$-test, $p = 0.74$). Furthermore, in the two weeks prior to our surveys, maximum daily air temperatures were marginally significantly warmer in 2012 (18.6 C $\pm$ 4.04 SD) than in 2010 (16.07 C $\pm$ 3.34 SD; two-tailed $t$-test,

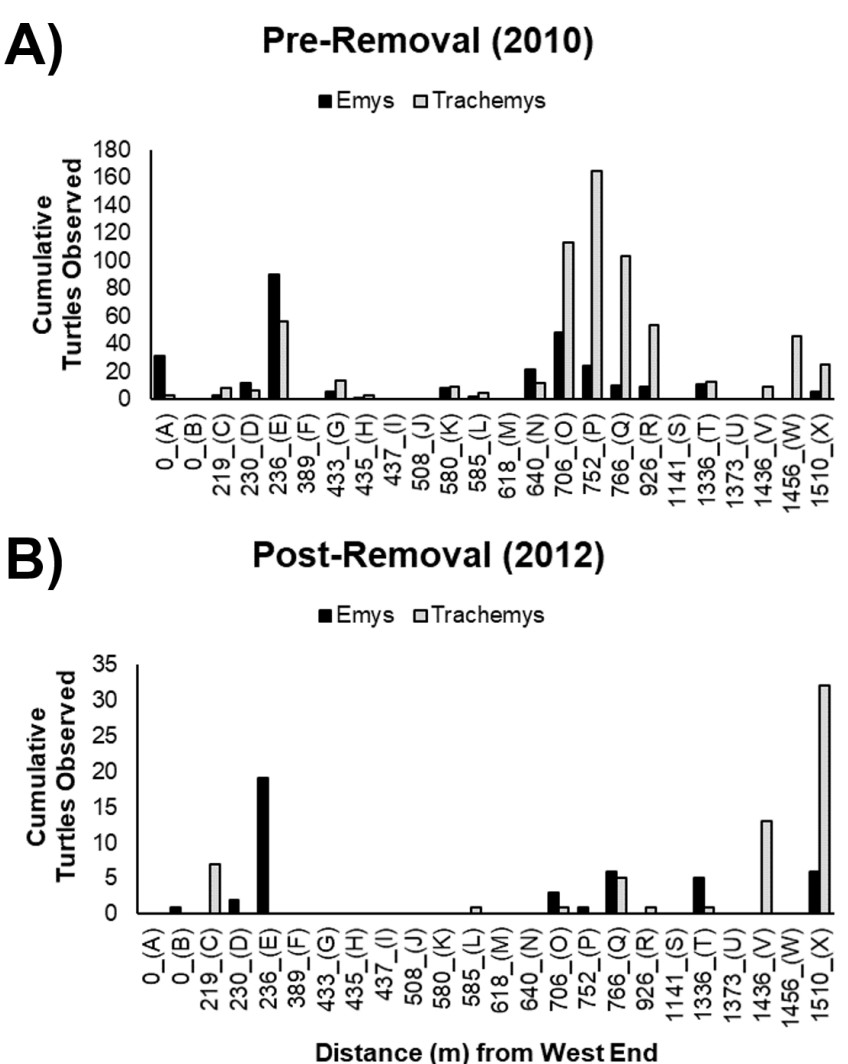

**Figure 3  Cumulative basking observations of native WPT (*Emys*) and introduced RES (*Trachemys*).** Basking observations before (A) and after (B) the RES removal are arrayed along a west-east gradient in the UCD Arboretum. Letters under the $x$-axis are basking site identifiers. Note the $y$-axes are on different scales in the (A, B).

$p = 0.08$). Air temperatures were also warmer in the winter (beginning of December to end of February) preceding the post-removal survey than the winter preceding the pre-removal survey (2009–2010, 13.4 C ± 3.35 SD; 2011–2012, 15.5 C ± 2.83 SD; two-tailed $t$-test, $p < 0.0001$). Colder water temperatures may thus have contributed to the lower overall turtle basking we observed in 2012, but this effect might have been modulated by warmer air temperatures prior to our 2012 surveys.

### Effects of RES removal on turtle basking

The interaction between removal treatment and distance from the west end of the UCD Arboretum was not significant ($p = 0.18$) and was removed from the model. Both treatment

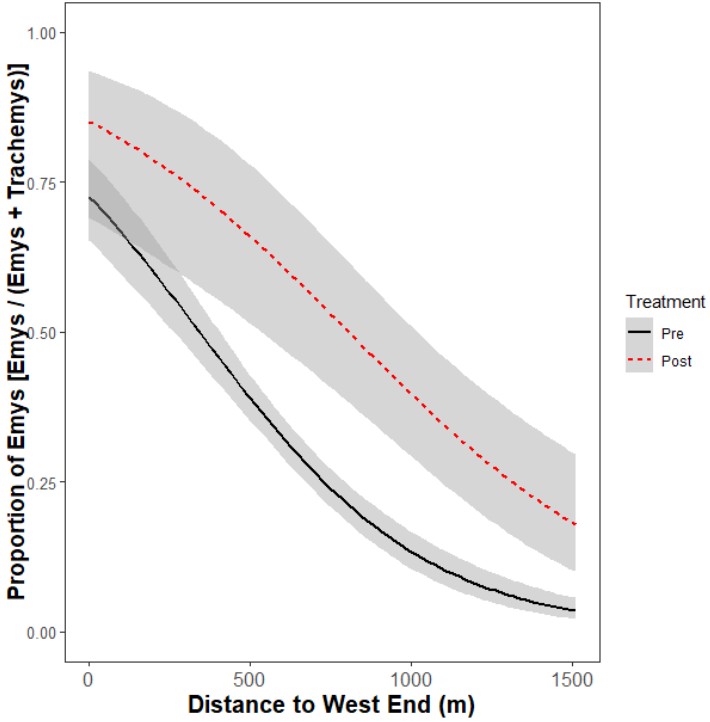

**Figure 4 Relative basking distribution of native WPT (Emys) to introduced RES (Trachemys) basking across the UCD Arboretum waterway pre- and post-removal.** Curves are the modeled ratios of WPT to RES basking along a west-east gradient in the UCD Arboretum pre- and post-removal (black and red curves, respectively). Models are for the relative basking distribution of the two species and account for the binary nature of these data and variation in sample sizes across basking sites and survey dates. The ratio of WPT to RES basking along the waterway was similarly WPT-biased in the west and RES-biased in the east in both years. WPT basking observations were higher after the RES removal.

($p < 0.0001$) and distance from the west end ($p < 0.0001$) were retained ($cR^2 = 0.31$, $mR^2 = 0.31$).

The non-significant interaction indicates the removal did not change the ratio basking turtles that were WPT across the UCD Arboretum. Both pre- and post-removal, the basking distribution of turtles was WPT-biased in the west end and RES-biased in the east end (Fig. 4). However, the proportion of basking individuals that were WPT increased from 30.5% pre-removal to 41.3% post-removal ($p < 0.0001$; Tukey's post-hoc test, function 'glht', package "multcomp"). Individual binomial GLMMs for each basking site showed removal treatment effects on the WPT:RES basking ratio for site Q ($p = 0.002$, 9% WPT to 55% WPT; Fig. 3) and a marginal effect for site O ($p = 0.09$, 30% WPT to 75% WPT; Fig. 3). All other individual basking sites showed no differences (all $p > 0.1$).

Pre-removal, the WPT basking distribution declined from west to east, and post-removal WPT basking had a relatively flat distribution (Fig. 5). We detected a shift in the absolute basking distribution of WPT with a significant interaction between removal treatment and distance from the west end (Poisson GLMM, $p = 0.012$, $cR^2 = 0.23$, $mR^2 = 0.06$). Individual GLMMs for each year indicated that distance from the west end was significantly associated

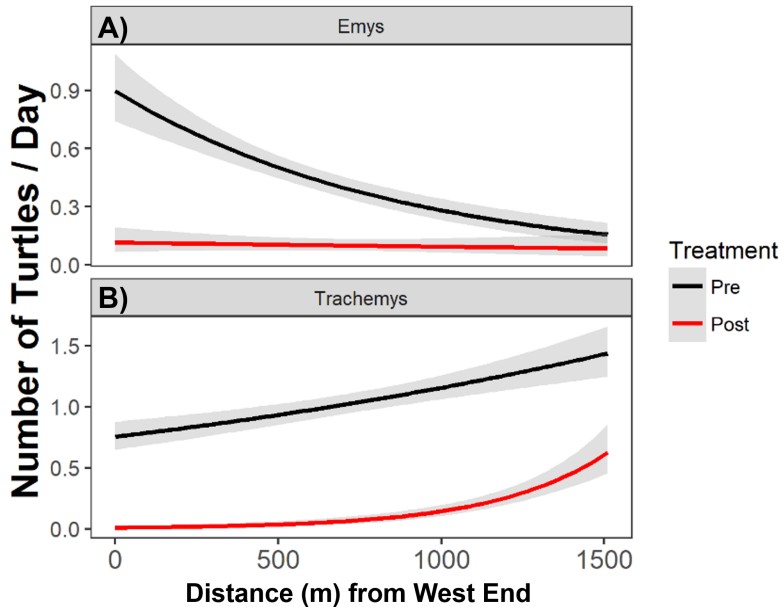

**Figure 5  The total number of native WPT (*Emys*) and introduced RES (*Trachemys*) basking along the UCD Arboretum.** Curves are the modeled daily number of WPT and RES along a west-east gradient in the UCD Arboretum pre- and post-removal (black and red curves, respectively). WPT (A) displayed a more even basking distribution after the RES removal, and RES (B) concentrated basking activity towards the east end of the Arboretum after most of their population was removed.

with WPT basking abundance in the pre-removal year ($p < 0.012$, $cR^2 = 0.27$, $mR^2 = 0.03$) but not in the post-removal year ($p = 0.55$).

WPT predominantly used the same basking sites post-removal but showed a more even distribution across basking sites, with more basking activity at two center-east sites (Q and X) compared to before the RES removal (Figs. 3A, 3B). Contingency table analyses showed that sites Q ($p = 0.01$) and X ($p = 0.001$) encompassed larger proportions of total WPT basking observations post-removal than pre-removal (Figs. 3A, 3B). All other sites made up similar proportions pre- and post-removal (all $p > 0.1$), though some sites had generally low basking activity (Figs. 3A, 3B), possibly limiting our power to detect shifts.

Our experimental removal of RES was associated with flatter observed distribution of WPT. Even so, if more eastern sites that were dominated by RES pre-removal (e.g., sites O, P, Q, and R) are also preferred WPT basking locations, then WPT should have increased basking at these sites post-removal. We did not see this shift.

After removal, remaining RES were sparse throughout much of the UCD Arboretum and concentrated in the east end (Figs. 3A, 5). For RES, a Poisson GLMM indicated a significant interaction between treatment and distance to the west end ($p < 0.0001$, $cR^2 = 0.30$, $mR^2 = 0.16$). Individual GLMMs for each year showed a positive relationship between RES basking and the distance to the west end pre-removal ($p < 0.0001$, $cR^2 = 0.27$, $mR^2 = 0.02$) and post-removal ($p < 0.0001$, $cR^2 = 0.14$, $mR^2 = 0.14$). The distance of each basking site from the west end explained substantially more of the variation in RES basking abundance post-removal than pre-removal, indicating that remaining RES
concentrated in the east end more strongly after we removed most of the RES population (Figs. 3A, 5). Contingency table analyses indicated that sites E, O, P, and R (Fig. 3) comprised lower proportions of total RES basking observations after the removal and site X (Fig. 3) comprised a higher proportion (all $p < 0.05$). Site Q made up similar proportions of total RES observations in both years ($p = 0.14$).

The RES remaining post-removal abandoned several basking sites that they previously used heavily (particularly sites O and P; Fig. 3) and shifted towards the east end of the UCD Arboretum (e.g., site X). This result suggests that RES prefer habitat at this end of the waterway and that, prior to our experiment, RES densities were high enough for intraspecific competition to force many RES into less preferred areas of the waterway. Our previous work showed that RES basking activity was highest at sites with shallow slopes, deeper water adjacent to the site, a steel mesh (rather than concrete or dirt) substrate, and high human activity (*Lambert et al., 2013*). Post-removal, RES basking activity was highest at the two sites (V and X) that maximized this combination of variables based on 2010 surveys (Fig. 4B from *Lambert et al., 2013*).

## DISCUSSION

Our experimental removal dramatically altered both RES and total turtle density in the UCD Arboretum by eliminating over half of the turtles in the waterway. Given the high population density of RES and given that we likely removed the majority of the RES population, we consider our RES removal effort substantial enough to have exerted an effect on WPT if the two species compete for food or basking sites. Our removal experiment offers new insights into competition for basking habitats and food between introduced RES and native WPT, producing four key results.

First, the prevalence of basking turtles at our survey sites post-removal was about 15% of that pre-removal, and this reduction in basking observations was measured in both species. We have no evidence that the removal of RES negatively affected the WPT population size, and a follow-up trapping survey confirmed that the number of WPT present remained roughly constant. Rather, it appears that the overall lower density of turtles in the UCD Arboretum allowed many WPT to either shift their basking activity patterns, redistribute themselves to sites that we were not monitoring, or both. Environmental differences, including cooler water temperatures during our post-removal monitoring, may also explain the lower WPT basking numbers, although our previous results from the same site suggest that the water temperature during our basking surveys would support maximal WPT basking activity (*Lambert et al., 2013*). It seems unlikely that differences in observers between years would have impacted these findings.

Second, after removing RES, we found that WPT basking activity at our monitoring sites shifted but did not increase at sites previously dominated by RES. Thus, we did not find evidence of strong interspecific competition for those sites. Interspecific competition is greatest at higher densities and the effects of an introduced competitor can similarly manifest or become most pronounced when the introduced species is at high densities (*Gurnell et al., 2004*). Therefore, competition is presumably greatest at high densities of

RES (or turtles generally) and perhaps influenced by the relative densities of both species. While earlier laboratory and mesocosm experiments suggest introduced RES outcompete native turtles for basking sites and other resources (*Cadi & Joly, 2003*; *Polo-Cavia, Lopez & Martin, 2010*; *Pearson, Avery & Spotila, 2015*), our results suggest more subtle effects found in complex, natural communities that are poorly predicted by simplified mesocosm experiments (*Skelly, 2002*; *Winkler & Van Buskirk, 2012*).

Third, after removing most RES, remaining RES concentrated their basking at sites (V and X) consistent with their previously identified preferred habitat characteristics (*Lambert et al., 2013*), suggesting that high RES densities prior to our experimental removal produced strong intraspecific competition, forcing many RES to use less-preferred basking habitat.

Fourth, we found that removing RES led to an increase in WPT body condition, suggesting that these turtle species compete for food. Whether this reflects interference competition (direct interactions between the two species), exploitation competition (both species indirectly competing for overlapping food resources), or a combination of the two is unclear. Experimental work on RES and other native turtles suggests RES may behaviorally prevent native turtles from obtaining sufficient food (*Cadi & Joly, 2004*; *Polo-Cavia, Lopez & Martin, 2011*; *Pearson, Avery & Spotila, 2015*), and our experimental removal may have reduced such interference if it does exist in this population. However, we also removed a substantial portion of the overall turtle community thereby reducing the overall pressure on food resources in the system. While differences in trapping dates between the two years (earlier post-removal) may have influenced our results, we think such effects are limited. The absence of an interaction between sex and treatment indicates that male and female WPT responded similarly to RES removal and suggests that differences in trapping date did not influence our results because of gravid females. Additionally, later trapping pre-removal could have allowed WPT to gain more mass over the active season compared to earlier trapping post-removal, making it hypothetically more challenging to detect a positive effect of RES removal on WPT body condition. Because of this, our body condition results may be a conservative estimate of body condition increase. Regardless of the mechanism, the ca. 40 g average increase in body condition we detected is substantial given that all WPT in our analysis pre-removal weighed under 1,100 g. To our knowledge, this result represents the first evidence from wild populations that introduced RES compete with native turtles for food and that RES removal can lead to improved body condition of native turtles.

## Should we remove RES to benefit declining native turtles?

A recent summary of research goals for effective conservation of WPT (*Thomson, Wright & Shaffer, 2016*) identified the need for a clearer quantitative understanding of the impact of introduced RES. Controlling invasive species is a substantial commitment that rarely eliminates the entire population, particularly in situations with continual introductions (*Kikillus, Hare & Hartley, 2012*; *Gaeta et al., 2015*; *García-Díaz et al., 2017*). Removing 177 RES from the UCD Arboretum was an intensive effort requiring >2,000 person-hours of field work across 40 days. A similar level of effort would conservatively cost the California Department of Fish and Wildlife $26,000–$31,000 in Scientific Aid hourly wages (L Patterson, pers. comm., 2019). While our study suggests that removing RES does influence

native turtle basking ecology and feeding, the potential benefits with respect to short-term basking-site usage appear quantitatively modest. However, the substantial increase in WPT body condition during the year following the RES removal suggests that removing RES meaningfully increased resource availability for WPT. Whether these returns justify the effort may well depend on several variables, including RES abundance / density, attitudes of local human residents to introduced RES, disease risk (*Héritier et al., 2017*), other potential axes of competition (e.g., nesting sites), and additional aspects of ecosystem health.

Our results also provide evidence that RES introductions may affect native turtles simply by inflating turtle densities in general (regardless of species identity). Therefore, removing RES may not necessarily relieve native turtles from a dominant competitor but, rather, may relieve ecological or behavioral pressures associated with high turtle densities and could conceivably result in unexpected responses by native species. One such unexpected response here was the substantial decrease in overall WPT basking observations after we removed over half of the turtle community, a result that suggests a change in WPT behavior and habitat use that our experimental design, with fixed monitoring sites, failed to capture. Unlike many other freshwater turtles, WPT are aggressive baskers—threatening, biting, pushing, and ramming other turtles from basking sites—and prefer to bask alone or in low numbers (*Bury & Wolfheim, 1973*). Reduced turtle densities post-removal may thus have allowed WPT to occupy other basking habitats in lower numbers as is their preference. Additionally, higher WPT body condition post-removal was likely influenced by there simply being fewer turtles overall competing for food in the UCD Arboretum. Improved body condition may also be the result of WPT adopting preferred basking behaviors, thereby improving digestive efficiency and mass gain. Future studies, including dietary research, that include unmanipulated control sites, pre-removal surveys that span multiple years and account for year-to-year variation, as well as a design that tracks the behavior of native turtles pre- and post-RES removal (e.g., using GPS-enabled radio transmitters) may better elucidate these unexpected outcomes on native turtles. Overall, our analyses suggest WPT responded to RES removal in a manner consistent with interspecific competition for food but inconsistent with strong interspecific competition for basking habitats, implying that removing RES may well be an important management strategy in some situations.

Alternatively, the direct management of basking habitat may be a more generally tractable conservation activity for WPT (*Spinks et al., 2003*; *Thomson, Wright & Shaffer, 2016*). In human-modified waterways, removal of floating basking sites for flood control and aesthetics (*Spinks et al., 2003*) could exacerbate competition for basking sites. Emerging research suggests that experimentally-added floating logs are preferred by WPT compared to bank-side basking sites and are more heavily used by WPT than RES, especially when they are isolated from human activities (Cossman et al., unpublished data). Adding artificial basking sites that favor WPT, alone or in combination with RES population reduction, is a simple, comparatively inexpensive manipulation that should be explored in future field experiments.

## Study limitations

The primary limitations of our study center on interpreting our basking results. We expected to observe fewer basking RES in the second year of study due to our intense removal effort but did not expect a concomitant decline in WPT observations. It is possible that water temperature, other environmental variation, or unforeseen consequences of our manipulation resulted in reduced overall turtle basking activity, or (more likely to us) radical shifts in basking to new and unmonitored locations, after the RES removal. Unfortunately, we cannot confidently identify which factor(s) resulted in fewer WPT basking observations. Although we studied both basking and feeding, we also recognize that our experiment did not address other potentially important axes of competition that are important for the continued recruitment and persistence of WPT populations. While we employed a before-after comparative design, the use of unmanipulated control sites would have improved our ability to make stronger inferences in this study. We do not believe that the lower number of WPT basking observations confounds our results because our analyses of relative basking distribution differences between species and years can accommodate sample size differences. Additionally, our analyses found that residual RES shifted their basking in intuitive ways (i.e., towards sites with preferred characteristics), increasing confidence in our results. While field experiments offer more biological realism than experiments in captivity, that added complexity may also yield unexpected results, such as changing a focal species' behaviors or habitat use.

## CONCLUSIONS

We present the first *in situ* field manipulation testing for competition between non-native RES and native turtles. Consistent with expectations based on laboratory and mesocosm studies, RES removal increased WPT body condition and altered WPT basking activity. However, contrary to expectations, this change in basking was not consistent with strong competition between RES and WPT for individual basking sites in the UCD Arboretum. Our results offer evidence for intraspecific competition for food and basking sites at high RES densities, underscore the value of manipulative field experiments in studying biological invasions, and suggest that removing introduced RES could be considered a useful, albeit logistically challenging, tool for managing wild WPT in some contexts. We encourage other researchers to replicate our field-based experiment, perhaps using control sites or multiple years of pre-removal observations. These modifications to our protocol would improve the ability to interpret competition between RES and native turtles and the magnitude of behavioral shifts that occur when removals lead to changes in both relative and absolute turtle densities.

## ACKNOWLEDGEMENTS

This work began while all authors were at UC Davis. The UCD Arboretum staff, members of the Peter Moyle and Janet Foley labs, Nicholas Buckmaster, Lauren Cassidy, Jillian Howard, Anna Jordan, Brian Mahardja, Hilary Rollins, Bryce Sullivan, and Matthew Young provided invaluable assistance.

### Funding

This work was supported by National Science Foundation DEB grants (1257648 & 1457832) and The Northern California Herpetological Society. The funders had no role in study design, data collection and analysis, decision to publish, or preparation of the manuscript.

### Grant Disclosures

The following grant information was disclosed by the authors:
National Science Foundation DEB grants: 1257648, 1457832).
The Northern California Herpetological Society.

### Competing Interests

The authors declare there are no competing interests.

### Author Contributions

- Max R. Lambert conceived and designed the experiments, performed the experiments, analyzed the data, prepared figures and/or tables, authored or reviewed drafts of the paper, approved the final draft.
- Jennifer M. McKenzie, Robyn M. Screen, Adam G. Clause, Benjamin B. Johnson and Genevieve G. Mount conceived and designed the experiments, performed the experiments, authored or reviewed drafts of the paper, approved the final draft.
- H Bradley Shaffer and Gregory B. Pauly conceived and designed the experiments, performed the experiments, contributed reagents/materials/analysis tools, authored or reviewed drafts of the paper, approved the final draft.

### Animal Ethics

The following information was supplied relating to ethical approvals (i.e., approving body and any reference numbers):
    The UC Davis Institutional Animal Care and Use Committee approved this research (IACUC Protocols #15263 and #16227).

### Field Study Permissions

The following information was supplied relating to field study approvals (i.e., approving body and any reference numbers):
    The California Department of Fish and Wildlife approved turtle trapping (Scientific Collecting Permits #2480, #4307, and #11663).

### Data Availability

    The raw data are available in the Supplemental Files.

### Supplemental Information

Supplemental information for this article can be found online at http://dx.doi.org/10.7717/peerj.7444#supplemental-information.

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
