# Peer review of "Experimental removal of introduced slider turtles offers new insight into competition with a native, threatened turtle"

_PeerJ, doi:10.7717/peerj.7444_

## Round 0.1 · original submission · Minor Revisions

Both reviewers considered that your manuscript represents a generally well-written and sound scientific contribution. They have provided numerous suggestions for improvements, especially the need for additional details in several places. I agree with these comments and have a few of my own. One reviewer provided more detailed comments in an attached file; please include these comments in your response document. I have added an annotated pdf of your manuscript, with highlights to indicate words that may need changing and inserted comments to suggest an alternative and/or to explain my reasoning. You do not need to respond to the suggestions on my pdf unless you disagree.

Editor’s Comments

L79-80. At first, I thought you had erred by including food consumption rate in interference competition. The following sentence indicates that I was wrong, but I think it would be helpful to provide a bit more description of how feeding rate could be a form of interference.
L111ff. We need more information on the study site: latitude and longitude, type of water body, general habitat description.
L125. Why did you test for condition effects at different times in the two years? Do you have any evidence that condition would not vary between late May and July or September? I agree with the reviewers that the potential effect of time of year should be a point of discussion.
L150, Fig. 4 and elsewhere. Your measure of the relative use of basking sites is ambiguous. You describe it as a proportion but present it as a ratio. Proportions would be WPT/(WPT + RES) and RES/(WPT + RES). Please clarify in Methods, figure caption and y-axis label, and anywhere else the term is used.
L214-215 (and elsewhere in paragraph). Provide means and variation before statistics. SD is preferred to SE for descriptive data.
L254-261. These sentences belong in the Discussion. A reviewer also makes the point that you have an excessive amount of discussion (i.e., interpretation of results) within your Results section. In some cases, I can see that it is useful to provide some interpretation so that subsequent findings are clearer. In other cases, such as L254-261, this is definitely more appropriate for the Discussion section. Please go over the Results in detail, noting each case of interpretation and make a careful decision about whether the interpretation is truly needed in Results or can be moved.
L309ff. I think there should be discussion of the differences between years and possible seasonal effects.
L345. What are ‘aggressive baskers’? It is not clear if ‘prefer to bask alone or in small numbers’ is supposed to be a definition or simply an added characteristic.

Reviewer 1 ·

Basic reporting

The manuscript is well written and provides mostly sufficient background and context. I have made a few suggestions for improvement in the specific comments to the authors.

Experimental design

Research question is well designed and executed. There are a few points that need improvement and clarification (i.e., body condition index). I specified these concerns in the general comments to the author.

Validity of the findings

Although some results were unexpected, the authors do well to discuss their findings.

Additional comments

General comments: This is a well written paper that addresses an important conservation question regarding the most invasive freshwater turtle species in the world and its effects on a threatened species. I have the following comments that I think would improve the quality of the manuscript.
Line 57: I think the authors can spend a little more time on native and introduced distribution of RES. PeerJ is not a turtle oriented journal and the readership in wide so more info here could be useful. Perhaps talking about introduction to other continents and then introduction to “non-native” states of the US. Saying “central USA” I do not think is accurate enough.
Line 60-61: I feel that going into more details about the results of these studied won’t hurt.
Line 74: Elaborate on these basking structures you mention (Lambert et al. 2013).
Line 86-88: Please provide more explanation about why/how is this a closed system. Is it fenced off completely. Working with turtles for over a decade even in fenced off areas, I learned that their movement is quite underestimated.
The distribution of basking should be further explained instead of just citing a previously published paper.
Line 110: More information about the study site is needed. What do the authors mean by optimal habitat. It is hard for readers to visualize this without further description. How big is the area etc?
Did dip netting and hand capture occur over the same time span?
Line 128-132: More appropriate approach is to use residual values of linear regression of PL to body mass as a body condition index. It looks like that is what the authors did in the figure, but they did not appropriately explain it here. Also were the data normally distributed or log transformed etc.? I like that they used individual WPT as a random effect.
Line 133: Is the model fit better when you removed non-significant variables from the model?
Line 139: This sentence sounds like you were monitoring 16 days before you started a removal. Please clarify/rephrase.
Line 142-146: I am intrigued by the visual survey technique. What was the distance between the observer and the basking structures? Were the turtles by any means disturbed by the observer? Also, how long did it take to complete one survey? In addition, did the same surveyor conduct surveys pre and post removal? This may be important when it comes to the potential biases especially IDing the turtle basking.
Line 154-155: explain the variation west to east as it pertains to both species.
Line 157: with proportional data?
Line 158-159: was the same package and function used as in the previous paragraph. Please specify.
Line 160: what interaction? I would say and if the interaction between treatment and distance was significant….
Line 164-165: Is there any chance these can be labeled on the Figure? If that will make the figure messy then I suggest referencing the supplemental material where we can find more info on the sites.
Line 174: what do you mean by remarked? Does the mark fade so fast?
Line 186-188: You trapped and measured the turtles during different months for each treatment (pre and post removal). Do you think this could affect the measurements given for example female reproductive cycle or the times of highest activity/foraging? Even if those effects were minimal I think this needs to be addressed in the methods or discussion. Also, is it normal for males to be heavier than females?
Line 197-199: Can you please state what were the pre-removal estimates and what were the confidence intervals for both estimates?
Line 208: Was this one site used previously by RES?
Line 234-235, line 270-273, line 274-276, line 280-282: hard to visualize where these sites are. Figure 3 helps with this, maybe refer to them.
Line 289-298: this paragraph may be a good one to mention a possibility of observer bias (or the lack of) or perhaps the area was more heavily used by people that would cause turtle disturbance and affect observations and therefore the number of turtles observed.
Line 301-302: briefly tell us what those preferred characteristics are.
Line 305: Remove the comma before parentheses
Line 337-357: the authors appropriately discuss the potential reasons for these unexpected results. Adding potential for future projects such as dietary studies (especially given they euthanized RES and stomach contents are available) may be something they want to add.
Line 364: For Cossman et al. unpubl. will there be a report available through supplemental material or similar?
Line 388: I would remove “large-scale”

·

Basic reporting

The manuscript is well written using clear and unambiguous, professional english throughout.

The introduction is based in the literature and provides sufficient context for the study. The methods section should have additional citations better rooting the authors work into the published framework.

The article structure, figures and raw data are professional and with some minor changes will be publishable.

The manuscript is mostly self contained. In one or two locations additional description of methods will improve this aspect of the manuscript. See the attached review.

Experimental design

The work described is original, and seeks to fill a knowledge gap in how an invasive species impacts a native species in more natural settings than previous studies.

As described the authors performed the work to a high technical and ethical standard.

Methods mostly were described well. In a few locations additional information should be provided to better describe the methods.

Validity of the findings

The underlying data have been provided, appear to be robust and statistically sound.

The conclusion is well stated, linked to the original question and mostly limited to supporting results.

Some speculation is made in the manuscript and discussion of alternative explanations of their findings should be expanded.

Additional comments

Overall, this is a well written manuscript that addresses an important question about the interaction between red-eared slider turtles and native turtle species interactions. The work described is suggestive that RES can impact native species in large bodies of water. This work builds on a number of previous studies that addressed this question in mesocosm type experiments.

The manuscript can be improved by providing some additional description about methods and expanding the discussion to better consider alternative explanations of the findings.

---

## Round 0.2 · accepted · Accept

The authors have responded thoughtfully and thoroughly to the comments of the reviewers and myself. The manuscript is now ready for publication following a small number of corrections.

1) The corresponding author has agreed by email to replace SE with SD as a measure of variation in the paragraph on L250-257. This should also apply to L224, 226, and the measure (SD) should also be specified.

2) The corresponding author has indicated that he will make minor changes in the Methods, Results, and Figure 4 to make it clear whether the measure of relative abundance is a ratio or a proportion. That is, the reader needs to know if the abundance of Emys at the west end of the study site is about 75% of all turtles or 75% of the abundance of Trachemys. Note that Fig. 4 should also have the units (m) on the x-axis.